# Loss of polycystins suppresses deciliation via the activation of the centrosomal integrity pathway

Vasileios Gerakopoulos, Peter Ngo, Leonidas Tsiokas

**The primary cilium is a microtubule-based, antenna-like organelle housing several signaling pathways. It follows a cyclic pattern of assembly and deciliation (disassembly and/or shedding), as cells exit and re-enter the cell cycle, respectively. In general, primary cilia loss leads to kidney cystogenesis. However, in animal models of autosomal dominant polycystic kidney disease, a major disease caused by mutations in the *polycystin* genes (*Pkd1* or *Pkd2*), primary cilia ablation or acceleration of deciliation suppresses cystic growth, whereas deceleration of deciliation enhances cystogenesis. Here, we show that deciliation is delayed in the cystic epithelium of a mouse model of postnatal deletion of *Pkd1* and in *Pkd1*- or *Pkd2*-null cells in culture. Mechanistic experiments show that PKD1 depletion activates the centrosomal integrity/mitotic surveillance pathway involving 53BP1, USP28, and p53 leading to a delay in deciliation. Reduced deciliation rate causes prolonged activation of cilia-based signaling pathways that could promote cystic growth. Our study links polycystins to cilia dynamics, identifies cellular deciliation downstream of the centrosomal integrity pathway, and helps explain pro-cystic effects of primary cilia in autosomal dominant polycystic kidney disease.**

## Introduction

The primary cilium is a solitary, antenna-like organelle present virtually in all cell types of the human body. It consists of the axoneme, a microtubular structure sheathed by a specialized form of the plasma membrane and the basal body, a modified centrosome forming the base of the cilium (Haimo & Rosenbaum, 1981). Cilium assembly and maintenance are achieved by intraflagellar transport (IFT), mediated by the concerted action of anterograde kinesins (IFT-B complex) and retrograde dynein (IFT-A complex) (Ishikawa & Marshall, 2011; Kobayashi & Dynlacht, 2011; Nigg & Stearns, 2011; Sung & Leroux, 2013). Cilia reach their maximum length in quiescence (G0) and gradually disassemble or shed upon cell cycle reentry (Pugacheva et al, 2007; Liu et al, 2018; Mirvis et al,

2018, 2019; Wang & Dynlacht, 2018). Gradual ciliary disassembly and shedding is a complex process and although it has been linked to cell cycle progression in cultured cells, its biological role in the context of the whole animal has been unclear (Sanchez & Dynlacht, 2016; Breslow & Holland, 2019). Accelerated or reduced rates of ciliary disassembly can decrease or increase the time cells spend in G1 and, thus, enhance or slow down cell cycle reentry, respectively (Kim et al, 2011; Li et al, 2011; Phua et al, 2017). There is also emerging evidence connecting cilia and the DNA damage response (Chaki et al, 2012; Johnson & Collis, 2016; Walz, 2017). Therefore, it is also conceivable that reduced disassembly rates could also trigger activation of the DNA damage response causing cell arrest in G2 and G1 to fully resorb cilia. In this case, failure to timely disassemble cilia could result in aneuploidy and genomic instability that could lead to uncontrolled proliferation. Therefore, changes in the rate of ciliary disassembly and/or shedding could have positive and negative effects on cell cycle progression dependent upon cell context and type and cilia-based signaling.

HEF1/AURKA/HDAC6 (Pugacheva et al, 2007), Nek2/Kif24 (Kim et al, 2015), and Plk1/Kif2A (Miyamoto et al, 2015) are major pathways of ciliary disassembly that are activated when cells are induced to reenter the cell cycle. In the HEF1/AURKA/HDAC6 pathway, HEF1 is a scaffold protein which when bound to AURKA induces AURKA autophosphorylation and activation (Nikonova et al, 2013). Activated AURKA phosphorylates HEF1 and HDAC6 (Nikonova et al, 2013). Activated HDAC6 leads to ciliary disassembly by de-acetylating $\alpha$-tubulin, inducing microtubule depolymerization (Pugacheva et al, 2007). This pathway is silent in G1/G0 and does not contribute to control of ciliary length at G1/G0, but is activated by up-regulation of HEF1 in response to mitogenic signaling (Pugacheva et al, 2007). AURKA and several other centrosomal proteins including NDE1, a negative regulator of ciliogenesis, form a large complex called the ciliary disassembly complex (CDC) (Gabriel et al, 2016). The activity of CDC can be positively or negatively modulated by several factors/pathways. These pathways include $Ca^{2+}$ signaling (Plotnikova et al, 2010, 2012), non-canonical Wnt5a signaling (Lee et al, 2012), changes in the composition of phospholipids of the ciliary membrane (Bielas et al, 2009; Phua et al, 2017), and others. Tumor suppressors such as Von Hippel–Lindau (pVHL) (Xu et al, 2010) and phosphatase and tensin homolog (PTEN) (Shnitsar et al, 2015) negatively regulate

Department of Cell Biology, University of Oklahoma Health Sciences Center, Oklahoma City, OK, USA

Correspondence: ltsiokas@ouhsc.edu

ciliary disassembly. In summary, ciliary disassembly is largely controlled by three pathways, which can be further influenced by extracellular/environmental and intracellular factors.

ADPKD is characterized by the formation of massive cysts in the kidney and increased cell proliferation (Bhunia et al, 2002; Kim et al, 2004; Zhou, 2009). In cell culture, loss of *Pkd1* or *Pkd2* seems to accelerate the G1 to S transition by impacting on levels of p21 (Bhunia et al, 2002; Kim et al, 2004; Li et al, 2005; Low et al, 2006). Consistently, the number of actively cycling cells in the cystic epithelium is always markedly increased in *Pkd1*- or *Pkd2*-null kidneys compared to wild-type kidneys (Zhou, 2009). Despite the increased proliferation of cystic cells, however, there are no gross ciliary abnormalities in animal models of ADPKD or tissues from patients with ADPKD, with the notable exception of longer cilia in a slowly progressing, hypomorphic mouse model of a human pathogenic mutation (Hopp et al, 2012). This observation suggests that highly proliferating cystic cells seem to have lost the ability to maintain coordination of cell cycle progression and ciliary turnover. These data are also consistent with the idea that persistent cilia-based mitogenic signaling may support cell proliferation and cyst expansion in cells lacking *Pkd1* or *Pkd2*. In fact, double-mutant mice lacking one of the *polycystin* genes and a gene essential for cilia formation, either *ift20* or *Kif3a*, show a much less severe phenotype, in terms of cyst formation and cell proliferation, compared with single mutants (Ma et al, 2013). Consistently, acceleration of cilia disassembly (Nikonova et al, 2018) suppresses cystic growth and improves kidney function, whereas deceleration of ciliary disassembly has the opposite effects in mouse models of ADPKD (Nikonova et al, 2014).

Our studies presented here show that deletion of *Pkd1* or *Pkd2* in several cell lines and MEFs leads to an inhibition of serum-induced ciliary disassembly and/or shedding and persistent activation of cilia-based signaling. Delayed disassembly is also seen in a postnatal mouse model of ADPKD. Delayed disassembly induced by the loss of *Pkd1* is secondary to the activation of the centrosomal integrity/mitotic surveillance (CI/MS) pathway involving the 53BP1-USP28-p53 axis.

## Results

### Deletion of *Pkd1/2* induces delayed cilia disassembly

We generated and characterized a mouse model of ADPKD using the tamoxifen-inducible *Ubc-Cre*ERT2 driver to postnatally delete the *Pkd1* gene globally (Figs 1A and S1A and B). As reported previously (Piontek et al, 2007; Ma et al, 2013), cell proliferation was markedly increased in cystic kidneys and the number of EdU-positive kidney epithelial cells was higher in 21-d-old *Ubc-Cre*ERT2;*Pkd1*f/f mice induced by 4-hydroxytamoxifen (4-OHT), compared with wild-type littermates. In addition, we noticed that the number of EdU-positive cells with primary cilia was increased by threefold in mutant kidneys compared with wild-type kidneys (Fig 1B and C). To further test whether cystic cells had longer cilia in the S phase than the wild-type cells, kidney sections were double-labeled for cilia and GEMININ, a protein that accumulates in the S phase (McGarry & Kirschner, 1998). GEMININ-positive cells with or without cilia were

very rare in wild-type kidneys. However, GEMININ-positive cells with cilia were easily identifiable in *Pkd1*-null kidneys (Fig S1C). These data raised the possibility that disassembly might be compromised or severely delayed in mutant cells.

Ciliary disassembly or shedding is a dynamic process difficult to be recapitulated in vivo. Therefore, we directly tested for an effect of the deletion of *Pkd1* or *Pkd2* on serum-induced deciliation in cell culture. Because cilia loss/shortening in response to serum can be mediated by gradual ciliary resorption/disassembly (Pugacheva et al, 2007), instant severing, and/or shedding (Mirvis et al, 2019), we scored cell cultures based on the presence or absence of detectable cilia to account for all modes of cilia loss. From here on, we adapt the term "deciliation" to include all forms of cilia loss. We used three different cell types: MEFs, NIH3T3 fibroblasts, and mouse renal epithelial cells (mIMCD3). Deletion of *Pkd1* or *Pkd2*, achieved either by homologous recombination in MEFs or CRISPR/Cas9 gene editing in NIH3T3 (Fig S2A and B) and mIMCD3 (Kleene & Kleene, 2016), did not affect the percentage of ciliated cells or ciliary length after 48 h of serum starvation (Fig S2C), suggesting a lack of an effect of the deletion of *Pkd1* or *Pkd2* on ciliary assembly. However, deletion of *Pkd1* or *Pkd2* significantly reduced serum-induced deciliation rates in all cell types, despite different kinetics among these cell types (Figs 2A–D and S2D–G).

To test whether these effects on deciliation were specific to PKD1 or PKD2 and not due to "off-target" effects of CRISPR/Cas9, we transfected back wild-type or mutant forms of PKD1 or PKD2. Adding-back human PKD1 or PKD2 corrected delayed deciliation, indicating a specific effect of PKD1 or PKD2 on deciliation. In contrast, transfection with PKD1S99I, a pathogenic mutant of PKD1 that fails to traffic properly to the plasma membrane (Kim et al, 2016), PKD2D511V, a pathogenic mutant of PKD2 (Reynolds et al, 1999; Koulen et al, 2002) or PKD2Y684del, which both show reduced channel activity (Zheng et al, 2018), failed to rescue delayed serum-induced deciliation (Fig 2E–H). These results suggested that the observed delay in cilia disassembly was specific to the absence of either polycystin and was reverted by adding back wild-type proteins. Furthermore, they supported the hypothesis that a fully functional PKD1/PKD2 complex, in terms of correct targeting to the plasma membrane and uncompromised channel activity, is essential for proper deciliation.

### p53 and TGFβ/Smad pathways are hyperactivated in *Pkd1*- and *Pkd2*-null cells

To obtain mechanistic insights of delayed deciliation and to also identify hyperactivated signaling pathways in *Pkd1*- or *Pkd2*-null cells, we screened a platform of 45 signaling pathways (Cignal Reporter Assays; QIAGEN) for their modulation by the inactivation of *Pkd1* or *Pkd2*. Although kinetics of cystic growth can be different in *Pkd1* versus *Pkd2* mutant kidneys, we focused on pathways that showed a similar direction of change in pathway activity in *Pkd1*- and *Pkd2*-null cells compared with wild type, despite different levels. Hyperactivation of p53 and TGFβ/Smad pathways stood out because they were hyperactivated in 10/10 independent experiments (Fig 3A) in NIH3T3 cells lacking PKD1 or PKD2 and mIMCD3 cells lacking PKD2. Other pathways were also up-regulated in mutant cells (i.e., ATF6, Hedgehog/Gli, E2F1, MYC, and TCF/LEF).

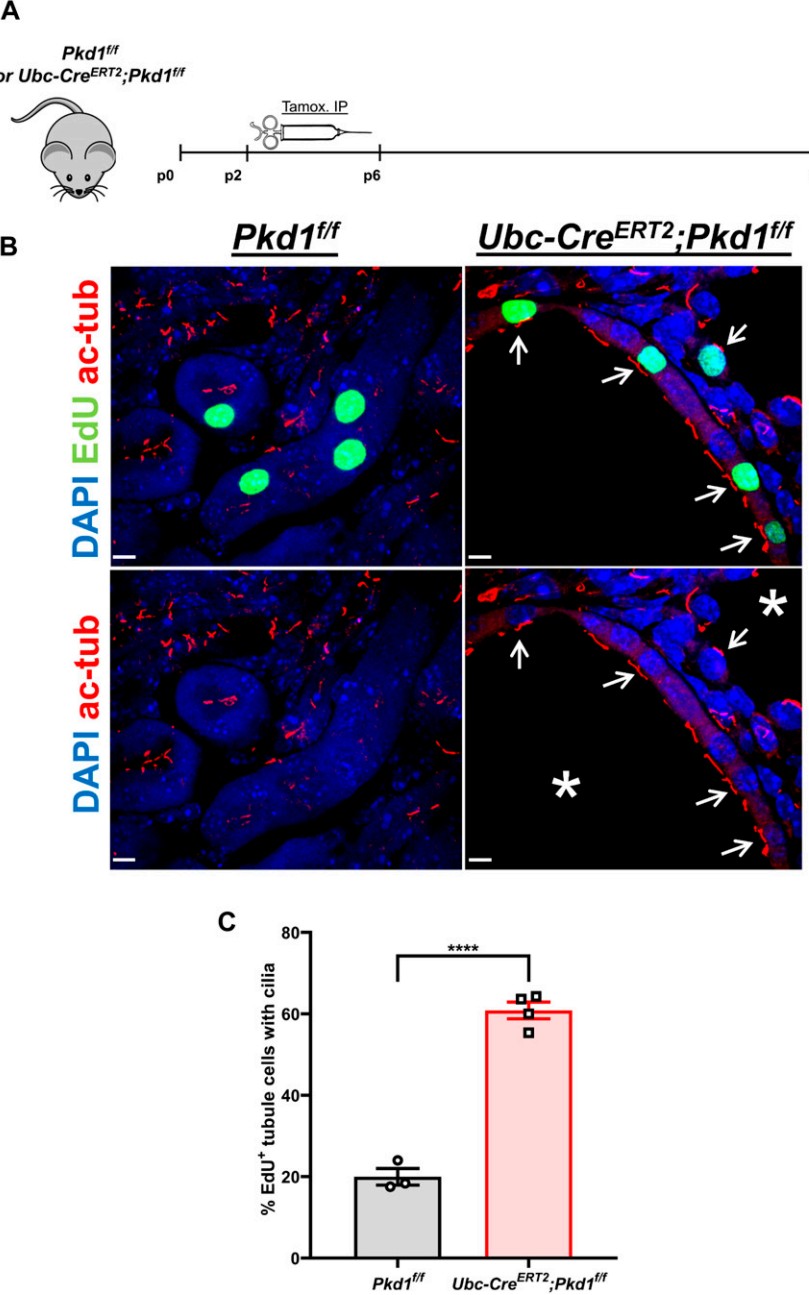

**A**

Pkd1^{f/f}
or Ubc-Cre^{ERT2};Pkd1^{f/f}

Tamox. IP

EdU, IP

p0  p2  p6  p20  p21

**B**

*Pkd1^{f/f}*          *Ubc-Cre^{ERT2};Pkd1^{f/f}*

DAPI EdU ac-tub

DAPI ac-tub

**C**

**Figure 1. Deletion of *Pkd1* increases the number of ciliated EdU⁺ cells in vivo.**
**(A)** Diagram showing administration of 4-hydroxytamoxifen (4-OHT) from P2 to P6 and intraperitoneal injection of EdU at P20. **(B)** Representative images of kidney sections stained for EdU (green) and acetylated α-tubulin (cilia, red) of P21 *Pkd1^{f/f}* or *Ubc-Cre^{ERT2};Pkd1^{f/f}* mice induced by 4-OHT from P2 to P6. Arrows indicate EdU⁺ cells with cilia. Asterisks indicate cysts. Scale bars: 5 μm. **(C)** Percent of EdU⁺ cells with cilia in *Pkd1^{f/f}* (n = 3) and *Ubc-Cre^{ERT2}; Pkd1^{f/f}* (n = 4) kidneys. 50–100 EdU⁺ cells per animal were scored for the presence of cilia. Data are presented as means ± SEM. *t* test, ****$P < 0.0001$.

However, pathways such as the MEF2 pathway responded differently to the deletion of *Pkd1* versus *Pkd2*. Deletion of *Pkd2* up-regulated, whereas deletion of *Pkd1* down-regulated MEF2 pathway activity, indicating that deletion of *Pkd1* or *Pkd2* did not completely overlap in pathway activity. Consistently, kidney-specific knockout of *Mef2c* reduced cystic growth in *Pkd2*-null kidneys (Xia et al, 2010). It would be interesting to examine cystic growth in *Pkd1/Mef2c* compound mice. The TGFβ/Smad2 pathway is known to be hyper-activated in kidneys lacking *Pkd1* (Hassane et al, 2010; Leonhard et al, 2016) and to induce kidney fibrosis, a prevalent feature of cystic kidneys. It is also known to promote proliferation in NIH3T3 cells

(Benzakour et al, 1992). Most relevant here, it requires an intact cilium for maximal activity (Clement et al, 2013). Therefore, we examined whether it showed prolonged activity in response to serum in *Pkd1*-null NIH3T3 cells compared with wild-type cells. Levels of activated phospho-Smad2 (p-Smad2) were determined at several time points during serum-induced ciliary disassembly in wild-type or *Pkd1*-null NIH3T3 cells previously synchronized in G1/G0 (Fig 3B and C). After serum starvation, Smad2 was not activated in wild-type or mutant cells indicating the absence of endogenously secreted ligands of the TGF superfamily. At 1 h after serum re-addition, the level of Smad2 activation was similar in both cell

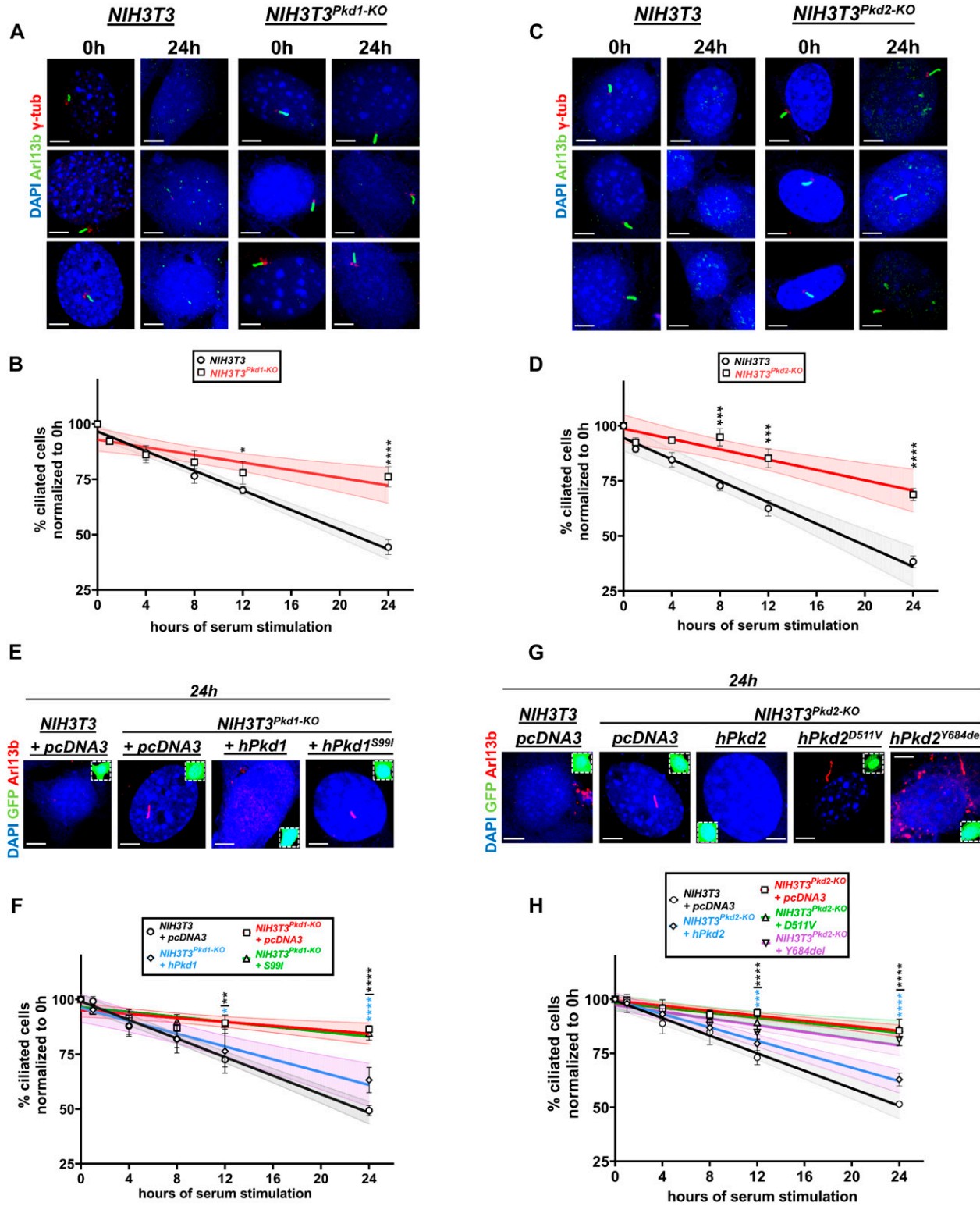

**Figure 2. Deletion of *Pkd1* or *Pkd2* decreases the rate of cell deciliation.**
**(A, B, C, D)** Representative images of wild-type or *Pkd1-null* NIH3T3 cells (A) and wild-type or *Pkd2-null* NIH3T3 cells (C) at 0 or 24 h after serum restimulation, stained for Arl13b (green) and γ-tubulin (red) to visualize ciliary axoneme and basal body, respectively. Serum-induced deciliation rates shown as best-fits with 95% confidence limits in indicated cell types (n = 3) (B, D). Scale bars: 5 μm. Data are presented as means ± SEM. Two-way ANOVA with Holm–Sidak's multiple comparisons test. *P < 0.05, ***P < 0.001, ****P < 0.0001. **(E, F, G, H)** Representative images of wild-type, *Pkd1-*, or *Pkd2-*null NIH3T3 cells transfected with the indicated constructs and GFP at 24 h after serum re-addition (E, G). GFP⁺ cells (green, insets) were evaluated for the presence or primary cilia via Arl13b (red) staining. Serum-induced deciliation rates shown as

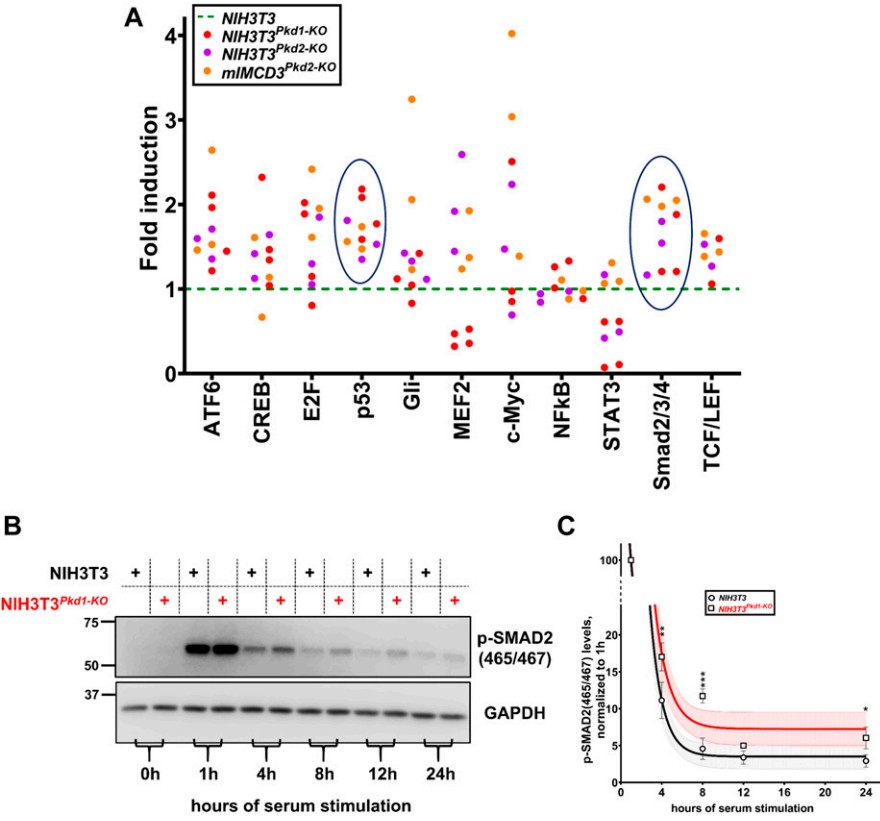

**Figure 3. Elevated p53 and TGFβ/Smad signaling in *Pkd1*-null NIH3T3 cells.**

**(A)** Summary data of a functional screen for signaling pathways modulated by *Pkd1* or *Pkd2* inactivation. Fold-induction over wild-type cells (shown by green dotted line) of the indicated signaling pathways in *NIH3T3^Pkd1-KO^* (n = 4), *NIH3T3^Pkd2-KO^* (n = 3), or *mIMCD3^Pkd2-KO^* (n = 3). TGFβ/Smad and p53 pathways (circled) were consistently induced in all 10 independent experiments. **(B)** Representative time-course of Smad2 activation (phospho-SMAD2 at 465 and 467) in response to serum re-addition in wild-type and *Pkd1*-null NIH3T3 cells. **(C)** Summary data of (B), shown as best-fits with 95% confidence limits in indicated cell types (n = 3). Data are presented as means ± SEM. Two-way ANOVA with Holm–Sidak's multiple comparisons test. *P < 0.05, **P < 0.01, ***P < 0.001.

Source data are available for this figure.

types. However, at all subsequent time points, p-Smad2 levels were higher in mutant cells, suggesting persistent activation of Smad2 during disassembly. These data are consistent with reduced deciliation rates of *Pkd1*-null cells.

### Inhibition of p53 rescues delayed deciliation in *Pkd1*- or *Pkd2*-null NIH3T3 cells

Signaling pathway analysis identified p53 as one of the pathways hyperactivated in *Pkd1* or *Pkd2* mutant cells. We confirmed that expression levels of p53 were higher in *Pkd1*- or *Pkd2*-null cells than those in wild-type cells (Fig 4A). Next, we determined whether p53 levels were also up-regulated in cystic kidneys in 21-d-old *Ksp-Cre^ERT2^;Pkd1^f/f^* and in 16- or 21-d-old *Ubc-Cre^ERT2^;Pkd1^f/f^* mice. We first characterized the *Ksp-Cre^ERT2^;Pkd1^f/f^* mouse model (Fig S3A and B). Whereas *Ubc-Cre* drives the expression of the Cre recombinase in all cell types, the *Ksp-Cre^ERT2^* uses the *Cdh16* promoter which is active in the epithelial cells of developing nephrons, the ureteric bud, and mesonephric tubules with low recombination efficiency in the proximal tubules (Shao et al, 2002). In the adult kidney, this promoter is active in the collecting ducts, loops of Henle and distal tubules, and to proximal tubules (Shao et al, 2002; Shibazaki et al, 2008). This driver is tamoxifen (4-OHT) inducible. In support of our

data in cell culture, protein expression levels of p53 in lysates of P21 cystic kidneys of *Ksp-Cre^ERT2^;Pkd1^f/f^* and P16 or P21 *Ubc-Cre^ERT2^; Pkd1^f/f^* mice, induced by 4-OHT from P2-P6, were increased in mutant kidneys (Figs 4B–D and S3C–E), suggesting that deletion of *Pkd1* either ubiquitously or specifically in kidney epithelial cells results in massive activation of p53 in cystic kidneys.

Because PKD1 and PKD2 are present in primary cilia (Yoder et al, 2002; Pazour et al, 2002b) and cyst formation and growth induced by the deletion of *Pkd1* or *Pkd2* share several characteristics of cystogenesis in cilia mutants (Davenport et al, 2007; Piontek et al, 2007), we reasoned that loss of *Pkd1* or *Pkd2* may have resulted in some sort of a structural defect of cilia and/or centrosome/basal body that could be sensed by the centrosomal integrity/mitotic surveillance pathway leading to the activation of p53. This pathway is activated by centrosomal loss and not by centrosomal amplification or DNA damage (Fong et al, 2016; Lambrus et al, 2016; Meitinger et al, 2016; Lambrus & Holland, 2017). Consistent with this hypothesis, we did not see differences in the number of phospho-H2AX (γ-H2AX)–positive wild-type and *Pkd1*-null NIH3T3 cells or wild-type and mutant kidneys staining (Fig S4A and B), indicating that DNA double-stranded breaks and genomic instability is unlikely to account for the delayed disassembly in mutant cells. Core components of the CI/MS pathway include 53BP1, USP28, and p53.

---

best-fits with 95% confidence limits in indicated cell types (n = 3) (F, H). Scale bars: 5 μm. Data are presented as means ± SEM. Two-way ANOVA with Holm–Sidak's multiple comparisons test. All comparisons were performed against the control group time points (*NIH3T3^Pkd1-KO^* transfected with pcDNA3 or *NIH3T3^Pkd2-KO^* transfected with pcDNA3, red curves). Color of the asterisks indicates comparison of the respective color group with the control group. *P < 0.05, **P < 0.01, ***P < 0.001, ****P < 0.0001.

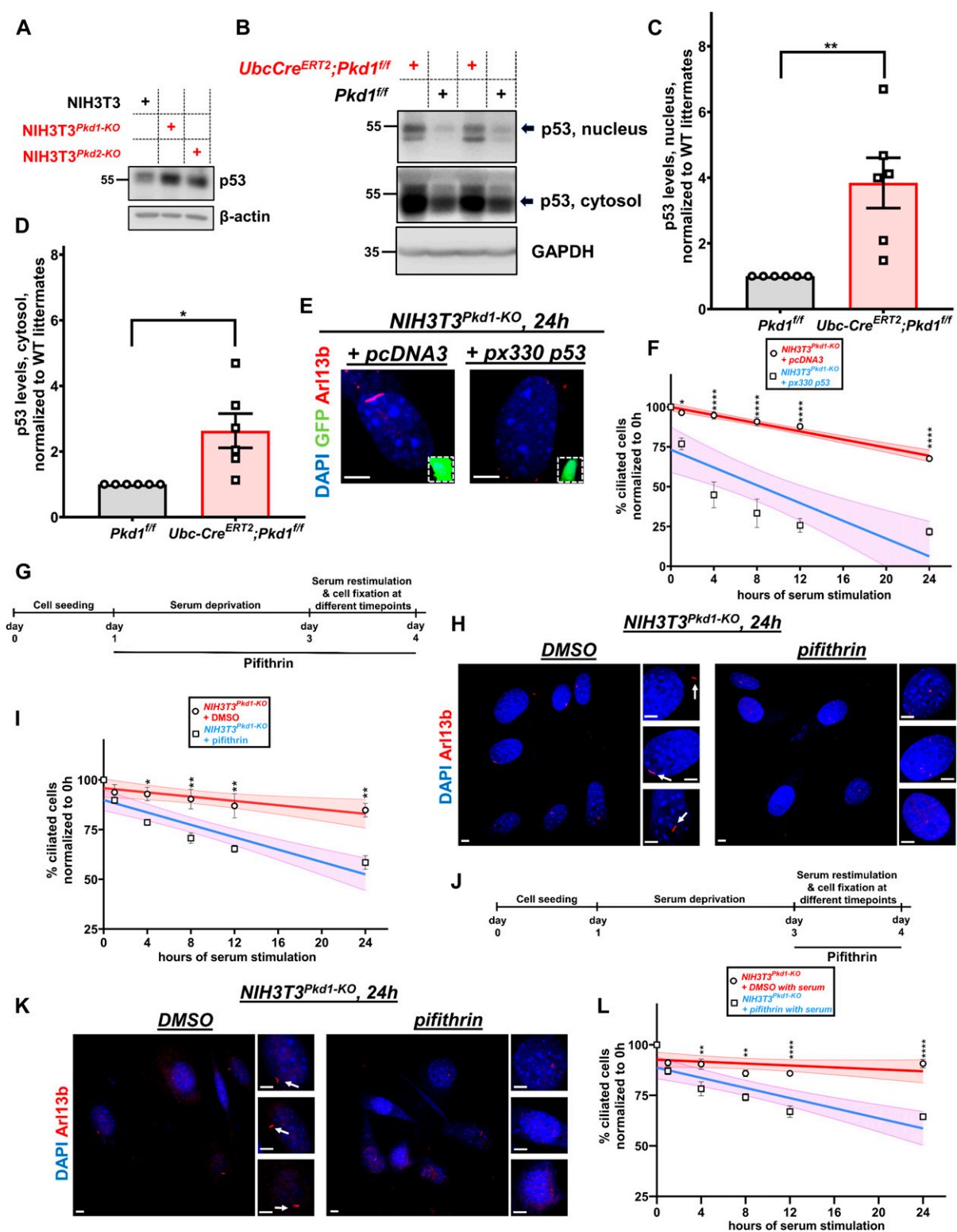

**Figure 4. Serum-induced deciliation is dependent on p53 in *Pkd1*-null NIH3T3 cells.**
**(A)** Expression levels of p53 in total cell lysates of wild-type, *Pkd1*-null, or *Pkd2*-null NIH3T3 cells. **(B)** Expression levels of p53 in nuclear and cytosolic extracts derived from P21 kidneys of *Pkd1*^f/f^ and *Ubc-Cre*^ERT2^*;Pkd1*^f/f^ mice treated with 4-OHT. **(C, D)** Quantification of protein levels of p53 in the nucleus (C) and cytosol (D), normalized to wild-type littermates pooled from three pairs of P16 and three pairs of P21 littermates. Data are presented as means ± SEM. *t* test. *$P < 0.05$, **$P < 0.01$. **(E, F)** Representative images from *Pkd1*-null NIH3T3 cells (E) co-transfected with the indicated constructs and GFP at 24 h after serum re-addition. *pcDNA3* was used as negative control. *px330 p53* expresses a p53-specific single guide RNA. GFP⁺ cells (green, insets) were evaluated for the presence or primary cilia via Arl13b (red) staining. Serum-induced

Inactivation of the *p53* gene via CRISPR/Cas9 gene editing (Figs 4E and F and S5A and B) or inhibition of its transcriptional activity by pifithrin (Figs 4G–I and S5C and D) restored delayed deciliation in *Pkd1*- or *Pkd2*-null NIH3T3 cells. Addition of pifithrin immediately after serum re-addition had similar effects in *Pkd1*- or *Pkd2*-null cells (Figs 4J–L and S5E and F), indicating an acute effect of p53 on deciliation within G1 and/or S phases of the same cycle. Deletion or inhibition of p53 had no effect on wild-type cells (Fig S5G–I). Thus, these data showed that deletion of *Pkd1* or *Pkd2* in cultured cells or kidneys leads to an increase in p53 levels and activity that is responsible for the inhibition of serum-induced cell deciliation.

Next, we tested whether depletion of 53BP1 and USP28 would have similar effects to the deletion of p53 on deciliation in *Pkd1*-null cells. Although not statistically significant, mRNA levels of both *53bp1* and *Usp28* were elevated in serum-starved *Pkd1*-null cells compared with wild-type cells (Fig 5A and B). This trend of increased mRNA levels of *Usp28* and *53bp1* at baseline (0 h) could be suggestive of activation of the CI/MS before serum re-addition. Down-regulation of both of these proteins (Fig S6A–D) restored deciliation in mutant cells (Fig 5C and D). Overall, these data suggested that deletion of *Pkd1* or *Pkd2* led to the activation of the CI/MS pathway, which in turn led to the activation of p53 activity to suppress deciliation. The elevated *53bp1* and *Usp28* mRNA levels in G1/G0 suggest that a centrosomal "defect" might have been induced earlier, but it did not manifest until cells were induced to deciliate. We speculated that overexpression of core components of the CI/MS pathway in wild-type NIH3T3 cells should be able to recapitulate the delay in deciliation observed in *Pkd1-null* cells. Indeed, over-expression of USP28, 53BP1, both USP28 and 53BP1, or p53 was able to suppress deciliation in wild-type cells (Fig 5E and F). Overall, these results support the conclusion that activation of the CI/MS pathway inhibits deciliation and not the opposite.

### *Pkd1*-null cells show decreased levels of NDE1 during serum-induced deciliation

To understand how increased p53 activity could inhibit deciliation, we examined expression levels of known disassembly factors or negative regulators of cilia formation. NDE1 is a negative regulator of ciliogenesis, whose expression is induced upon cell cycle reentry but suppressed in G1/G0 via SCF^FBW7 E3 ligase (Maskey et al, 2015). *Pkd1*-null cells showed reduced levels of NDE1 upon serum re-addition, compared with wild-type cells (Fig 6A and B). To further test the possible role of p53 in the reduction of NDE1 protein expression levels during deciliation, we treated wild-type NIH3T3 cells with nutlin, a stabilizer of p53, immediately after serum restimulation, and determined NDE1 protein levels at 24 h and the percentage of ciliated cells at 8 and 24 h after serum re-addition. Nutlin treatment compromised NDE1 protein levels and cell deciliation in wild-type NIH3T3 cells (Fig 6C–E). These data suggested that increased p53 activity can suppress NDE1 protein levels. However, because deciliation was suppressed at concentrations of nutlin that were ineffective in suppressing NDE1 levels (20 µM), we suggest that other p53-regulated proteins can be affected in parallel with NDE1 to suppress deciliation in these cells.

We reasoned that adding back NDE1 in *Pkd1*-null cells should promote deciliation, if low NDE1 levels had a causal role in reduced deciliation rate observed in these cells. However, overexpression of NDE1 throughout the cell cycle has a robust effect in suppressing ciliation. To maximize the expression of NDE1 in the G1/S phase (when deciliation occurs), we used a tetracycline-inducible vector to induce expression of NDE1 at the time of serum re-addition. We also fused at its C terminus amino acid residues 1–110 of human GEMININ containing a well-established APC/C phosphodegron (McGarry & Kirschner, 1998), to ensure almost complete degradation at G0 (Fig 6F). As shown in Fig 6G–I, the percentage of ciliated cells, at 12 and 24 h after serum re-addition, in *Pkd1*-null NIH3T3 cells transfected with NDE1–hGEM and induced by 2 µg/ml doxycycline at the 0-h time point, was reduced to wild-type levels. Overexpression of this construct and induction of expression by similar concentration of doxycycline did not accelerate disassembly in wild-type cells. We reasoned that because ciliation levels in mock-transfected and NDE1–hGEM–transfected *Pkd1*-null cells are similar at 0 h of serum re-addition, but deciliation is enhanced after serum re-addition in NDE1–hGEM–transfected mutant cells, NDE1 could function as a serum-induced deciliation factor, and its observed down-regulation in *Pkd1*-null cells could have a causal effect in the suppressed rate of deciliation in these cells. Overall, these data suggest that NDE1 could function downstream of PKD1 in serum-induced deciliation in NIH3T3 cells.

## Discussion

Although it was realized 18 yr ago that cilia can have a major role in the pathogenesis of polycystic kidney disease (Pazour & Rosenbaum, 2002; Pazour et al, 2002a), the exact cellular mechanism of how cilia contribute to cyst formation/progression in ADPKD still remains unknown. Because cilia appear of normal size in several mouse models of ADPKD, it is thought that loss of polycystins may have a primary role in modulating ciliary function rather than structure (Nauli et al, 2003; Pazour, 2004; Ma et al, 2013). Our studies challenge this idea, by showing that altered ciliary function is secondary to defective deciliation. This is consistent

---

deciliation rates are shown as best-fits with 95% confidence limits in indicated cell types (n = 3) (F). Scale bars: 5 µm. Data are presented as means ± SEM. Two-way ANOVA with Holm–Sidak's multiple comparisons test. \**P* < 0.05, \*\*\*\**P* < 0.0001. **(G, H, I)** Diagram showing treatment with 10 µM pifithrin during serum starvation and during serum restimulation (G). Representative images of primary cilia (Arl13b, red) from *Pkd1*-null cells (H) at 24 h after serum restimulation, after the indicated treatment. Serum-induced deciliation rates are shown as best-fits with 95% confidence limits in indicated cell types (n = 3) (I). Arrows indicate primary cilia. Scale bars: 5 µm. Data are presented as means ± SEM. Two-way ANOVA with Holm-Sidak's multiple comparisons test. \**P* < 0.05, \*\**P* < 0.01. **(J, K, L)** Diagram showing treatment with 10 µM pifithrin only during serum restimulation (J). Representative images of primary cilia (Arl13b, red) from *Pkd1*-null cells (K) at 24 h after serum restimulation, after the indicated treatment. Serum-induced deciliation rates are shown as best-fits with 95% confidence limits in indicated cell types (n = 3) (L). Arrows indicate primary cilia. Scale bars: 5 µm. Data are presented as means ± SEM. Two-way ANOVA with Holm–Sidak's multiple comparisons test. \*\**P* < 0.01, \*\*\*\**P* < 0.0001.
Source data are available for this figure.

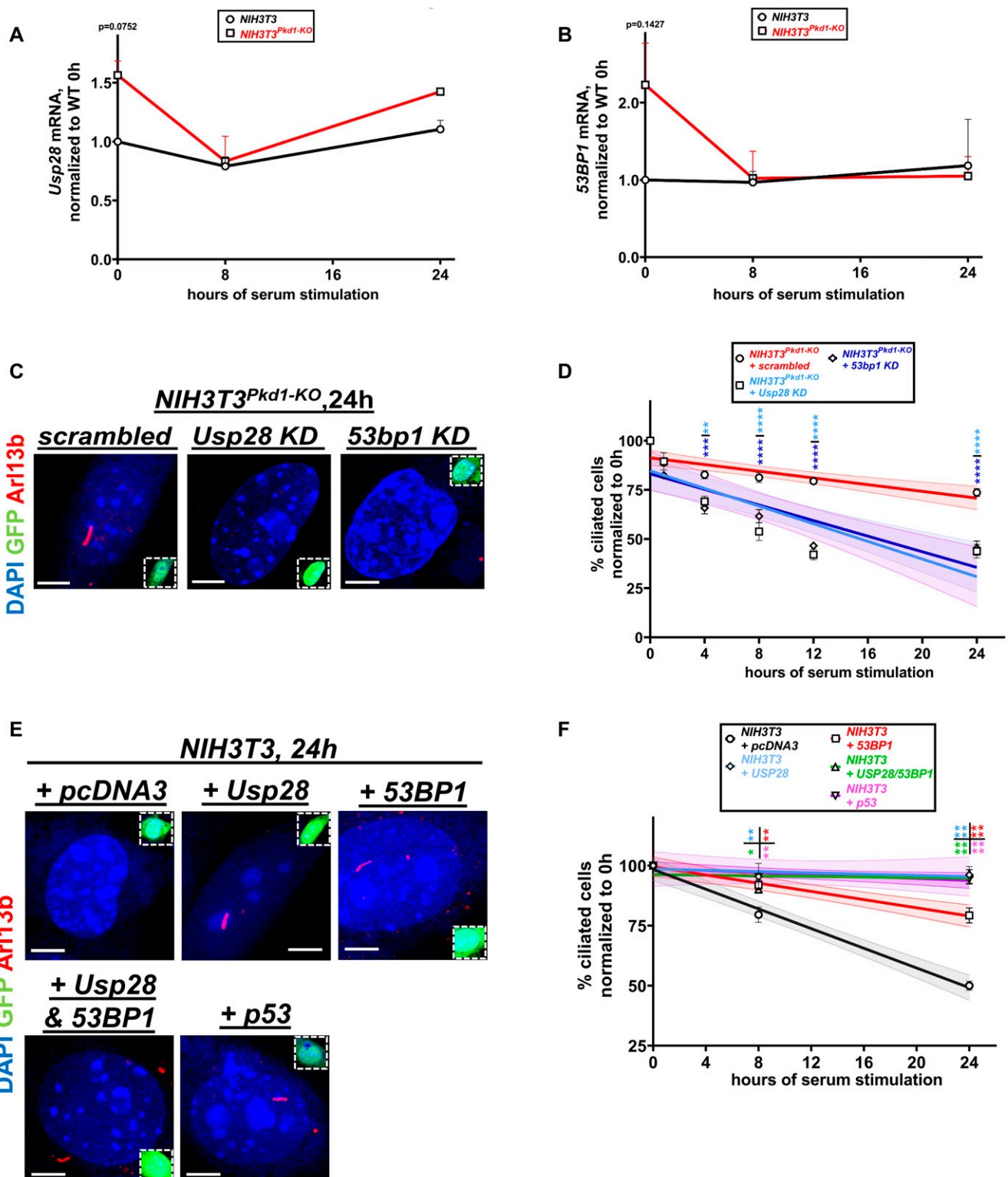

**Figure 5. Serum-induced deciliation rate is modulated by components of the centrosomal integrity/mitotic surveillance pathway.**
**(A, B)** *Usp28* (A) and *53bp1* (B) mRNA levels in wild-type or *Pkd1-null* NIH3T3 cells, at various time points after serum re-addition (n = 3). Data are presented as means ± SEM. Two-way ANOVA with Holm–Sidak's multiple comparisons test. **(C, D)** Representative images from *Pkd1*-null cells transfected with the indicated constructs and GFP at 24 h after serum re-addition (C). GFP⁺ cells (green, insets) were evaluated for the presence or primary cilia via Arl13b (red) staining (C). Serum-induced deciliation rates are shown as best-fits with 95% confidence limits in indicated cell types (n = 3) (D). Scale bars: 5 μm. Data are presented as means ± SEM. Two-way ANOVA with Holm–Sidak's multiple comparisons test. All comparisons were performed against the control group time points (*NIH3T3^{Pkd1-KO}* transfected with scrambled siRNA, red

with previous data showing that primary cilia ablation (Ma et al, 2013) or acceleration of cilia disassembly suppresses cystic growth (Nikonova et al, 2018), whereas deceleration of ciliary disassembly enhances cystic growth in mouse models of ADPKD (Nikonova et al, 2014). Our data point to serum-induced deciliation as a cellular process impacted by the loss of polycystins.

We suggest that the CI/MS pathway is a critical intermediary between the loss of polycystins and reduced rate of serum-induced deciliation. This conclusion is based on the following lines of evidence. First, cells lacking *Pkd1* or *Pkd2* do not show differences in ciliation levels or ciliary length upon serum starvation, indicating that loss of PKD1 or PKD2 does not impact cilia assembly programs. In contrast, cells lacking *Pkd1* or *Pkd2* show a reduced rate of serum-induced deciliation that is rescued by knocking down or inhibiting components of the CI/MS pathway such as USP28, 53BP1, and p53. Conversely, overexpression of these factors in wild-type cells induces delayed deciliation, indicating that activation of the CI/MS pathway is downstream of polycystin loss and upstream of delayed disassembly. Second, renal cells lining the cystic epithelium in ADPKD mouse models show not only a higher number of EdU-positive cells but also a higher number of EdU-positive cells with primary cilia than the wild-type cells, suggesting that cell deciliation is compromised, yet entry into S is up-regulated in mutant cells. This suggests a possible uncoupling of the tight coordination of deciliation and cell cycle progression. Third, overexpression of NDE1 during disassembly induces deciliation in mutant cells, but not in wild-type cells. This finding indicates that NDE1 has a negative effect on cilia biogenesis and maintenance, but not on cilia undergoing disassembly or shedding, despite the fact that it was identified as a component of the CDC (Gabriel et al, 2016). An important question is how p53 activation leads to reduced protein levels of NDE1 expression. Because p53 mediates its effects by transcriptional activation, we believe that the effect of p53 on NDE1 is indirect, perhaps via transcriptional induction of factor(s) that down-regulate NDE1 and additional deciliation factors. These lines of evidence lead us to propose a model in which, deletion of polycystins activates the CI/MS pathway by increasing the levels/activity of USP28, 53BP1, and p53. Activation of this pathway pauses serum-induced deciliation allowing persistent activation of ciliary signaling pathways. These pathways can include the TGF-$\beta$/Smad, Hedgehog, canonical Wnt/$\beta$-catenin, and others that could promote cystic growth in a linear and/or parallel ways (Fig 7).

Our data, however, do not support a direct effect of the loss of polycystins on serum-induced deciliation but rather an indirect effect via the activation of the CI/MS pathway. We show that deletion or pharmacologic inhibition of p53, 53BP1, or USP28 restores deciliation rates in cells lacking *Pkd1*. These data suggest that these proteins function downstream of the loss of *Pkd1* in deciliation. We do not know, yet, exactly how loss of *Pkd1* or *Pkd2* activates the CI/

MS pathway. Because this pathway can be activated by centrosomal loss, mitotic delay, or both, in which case centrosomal loss induces mitotic delay (Lambrus & Holland, 2017), it is possible that one of the normal functions of the polycystins could be to maintain CI. However, we do not see centrosomal loss, gross centrosomal abnormalities, such as fragmentation, or amplification in *Pkd1*- or *Pkd2*-null cells, despite previous reports of centrosomal amplification in *Pkd1*- or *Pkd2*-null MEFs and kidneys (Battini et al, 2008; Burtey et al, 2008). It should be noted that even if centrosomal amplification occurs in *Pkd1*-null NIH3T3 cells above their known levels of 6% (Wong et al, 2015), the CI/MS pathway should not be activated by centrosomal amplification but rather by centrosomal loss. We favor the idea that wild-type polycystins may have an essential role in maintaining CI, and their loss induces subtle structural changes even before cells are induced to enter the cell cycle, because *53bp1* and *Usp28* mRNA levels were elevated in serum-starved cells. Because the centrosome is a membrane-less organelle, it is unlikely that the PKD1/PKD2 complex physically contributes to CI. However, PKD1/PKD2–mediated Ca$^{2+}$ signaling could affect critical aspects of centrosome biogenesis, maturation, and maintenance. In support of this hypothesis, mutant forms of PKD1 or PKD2 that compromise channel activity failed to rescue deciliation defects in *Pkd1*- or *Pkd2*-mutant cells, indicating that presence of a functional polycystin complex is necessary for CI/normal deciliation. More work in the future would be needed to identify the mechanism by which polycystin loss triggers the CI/MS pathway.

Our data show that the percentage of ciliated cells or ciliary length is unchanged in serum-starved cells lacking *Pkd1* or *Pkd2* and wild-type cells. Therefore, we do not have evidence to believe that loss of polycystins affects cilia assembly programs. This is consistent with our mouse data, in which ciliary length does not seem to be grossly different between wild-type and mutant cells, unless cells are actively cycling as determined by EdU labeling. It can be argued that a 24-h pulse of EdU could mark cells in all phases of the cell cycle, including G1. However, at P21, cell proliferation is extremely low in the kidney, as shown earlier that a 3-h BrdU pulse at P24 labeled only 4% of cystic cells and 0% of wild-type cells (Shibazaki et al, 2008). Therefore, the chance of an EdU-positive cell to be in the G1 phase after having gone through the S, G2, and M phases is quite low. Moreover, a cell that has transited from S to G1 within 24 h should produce two EdU-labeled daughter cells localized next to each other, which we do not see. Therefore, we favor the idea that EdU-positive cells represent cells mostly in the S phase and to a lesser extent in G2. This is supported by our data identifying GEMININ-positive cells with cilia in mutant kidneys, but not in wild-type kidneys. Based on DAPI staining, we do not see many dividing cells and according to γ-H2AX staining, we conclude that cells undergoing DNA damage or experiencing genomic

curve). Color of the asterisks indicates comparison of the respective color group with the control group. **$P < 0.01$, ***$P < 0.001$, ****$P < 0.0001$. **(E, F)** Representative images from wild-type NIH3T3 cells transfected with the indicated constructs and GFP at 24 h after serum re-addition. GFP$^+$ cells (green, insets) were evaluated for the presence or primary cilia via Arl13b (red) staining (E). Serum-induced deciliation rates are shown as best-fits with 95% confidence limits in indicated cell types (n = 3) (F). Scale bars: 5 $\mu$m. All comparisons were performed against the control group time points (NIH3T3 transfected with *pcDNA3*, black curve). Color of the asterisks indicates comparison of the respective color group with the control group. Data are presented as means ± SEM. Two-way ANOVA with Holm–Sidak's multiple comparisons test. *$P < 0.05$, **$P < 0.01$, ****$P < 0.0001$.

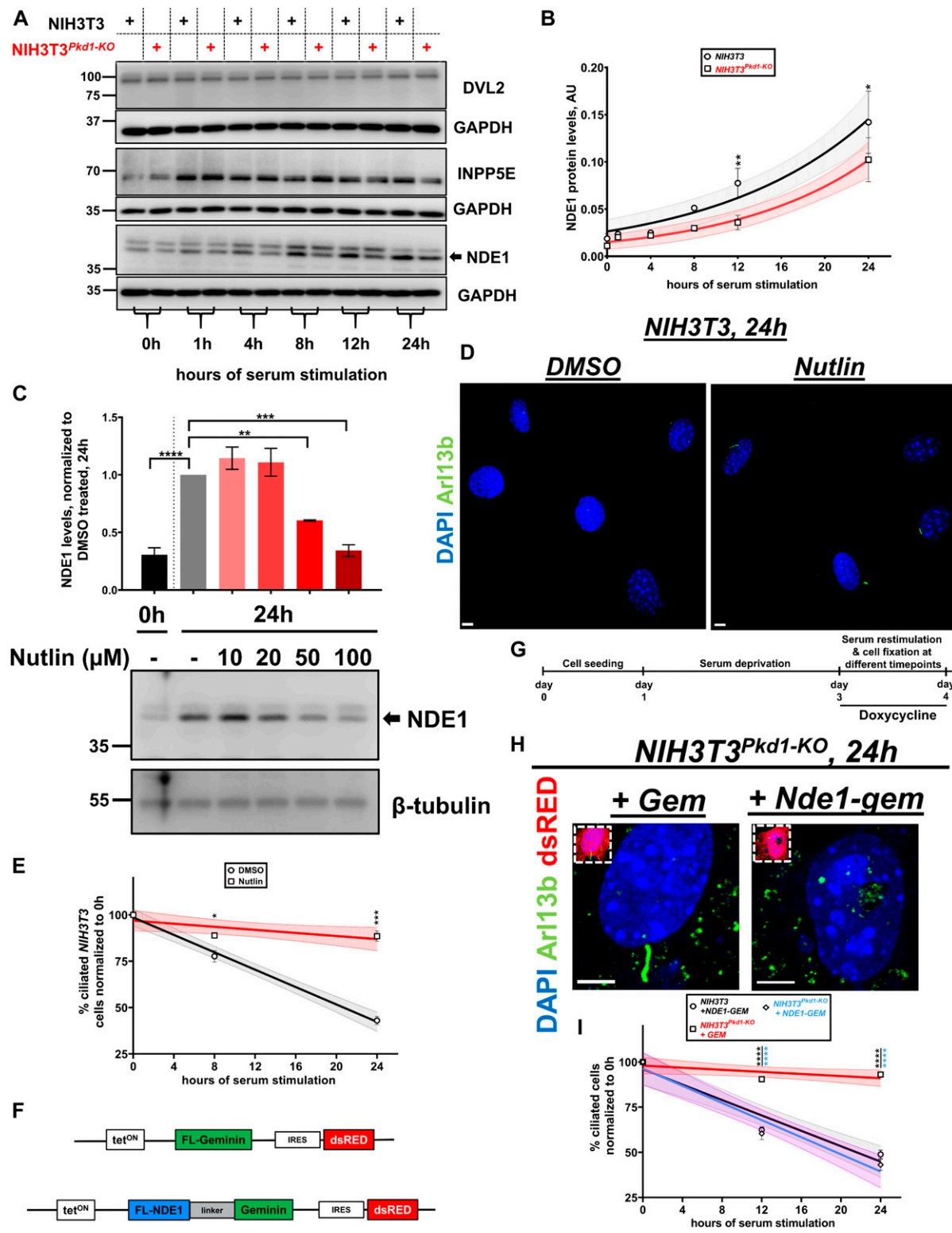

**Figure 6. p53 has a negative role in cilia disassembly via down-regulation of NDE1.**
**(A)** Representative time-course of protein levels of various disassembly factors during deciliation in response to serum re-addition in wild-type and *Pkd1*-null NIH3T3 cells. **(B)** Summary data of NDE1 protein levels in (A), shown as best-fits with 95% confidence limits in indicated cell types (n = 3). Data are presented as means ± SEM. Two-way ANOVA with Holm–Sidak's multiple comparisons test. *$P < 0.05$, **$P < 0.01$. **(C)** Bottom: representative protein levels of NDE1 in wild-type NIH3T3 cells at 0 or 24 h after serum stimulation, after the indicated treatments. Top: summary data of NDE1 levels, normalized to mock treated at 24 h (gray bar) (n = 3). Data are presented as means ± SEM. One-way ANOVA with Holm–Sidak's multiple comparisons test, **$P < 0.01$, ***$P < 0.001$, ****$P < 0.0001$. **(D, E)** Representative images from wild-type NIH3T3 cells at 24 h

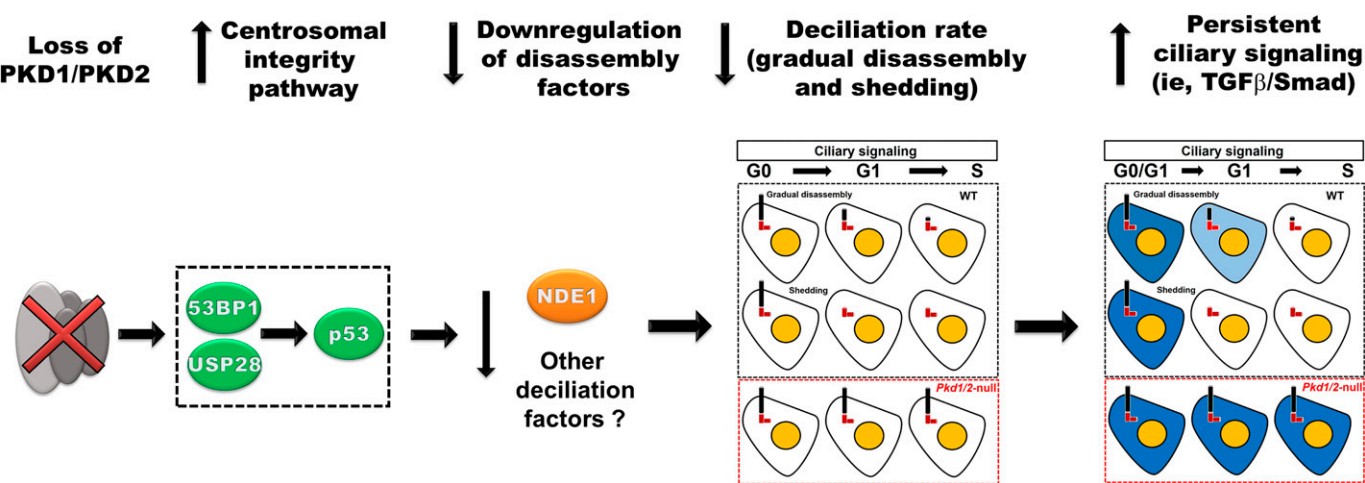

**Figure 7. Proposed mechanism by which depletion of PKD1/PKD2 leads to increased cilia-related signaling.**
Depletion of PKD1 or PKD2 induces the centrosomal integrity/mitotic surveillance pathway leading to reduced ciliary disassembly and/or shedding (deciliation) rate. Reduced deciliation rate prolongs activation of cilia-dependent signaling pathways (i.e., TGFβ/Smad, Hedgehog, and Wnt/βcatenin).

instability are barely detectable in either wild-type or mutant kidneys. Therefore, we propose that cystic cells progressed to the S phase with partially resorbed cilia or intact cilia. At first, this hypothesis would seem at odds with the increased overall cell proliferation in cystic kidneys. However, there could be several explanations that warrant further consideration. Others and we proposed that accelerated disassembly, or cilia shortening/loss, can speed up cell cycle reentry, whereas delayed disassembly or abnormally long cilia can have the opposite effects (Pan & Snell, 2007; Kim & Tsiokas, 2011; Sung & Li, 2011; Wang & Dynlacht, 2018). These data have suggested that the primary cilium may function as a physical checkpoint of cell cycle progression. Somewhat counterintuitive to this idea, though, is the fact that primary cilia house numerous signaling pathways, most of which have strong mitogenic effects (Schneider et al, 2005; Christensen et al, 2012; Yeh et al, 2013). In addition, cilia can promote or suppress Hedgehog signaling-dependent tumorigenesis depending on the driver mutation (Han et al, 2009; Wong et al, 2009). Finally, cilia confer chemoresistance to certain types of tumors (Jenks et al, 2018). Therefore, it appears that primary cilia can have both positive and negative effects on cell cycle progression. Although they are required for the initiation of mitogenic signaling, they also need to timely disassemble or shed for normal cell cycle progression. Assuming that delayed disassembly does not cause complete cell arrest, it could support persistent cilia-based mitogenic signaling in some number of cells that perhaps stochastically, would not normally enter the cell cycle.

This extra time that cells are exposed to a mitogenic stimulus could lead to more cells eventually entering the cell cycle, increasing the number of proliferating cells. Nevertheless, whether and how delayed deciliation can promote cell cycle reentry in the setting of ADPKD requires further study.

Overall, our studies put forth a new model in which a physiological cellular function of polycystins is to maintain CI. Loss of this function activates the CI/MS pathway, which in turn signals cells to slow down the rate of deciliation. As a result, cilia-based signaling can persist, promoting cystic growth (Fig 7). The molecular identity of cilia-based signaling pathway(s) with cyst promoting activity is still outstanding, but it could be pathways that mediate mitogenic signaling. Persistent activation of cilia-based mitogenic signaling can recruit kidney epithelial cells in G1/G0 to eventually enter the cell cycle, contributing to cyst expansion and growth.

# Materials and Methods

### Cell culture

Pkd1+/+ (wild-type) and Pkd1−/− MEFs were obtained from the PKD Baltimore Center. Pkd2+/+ and Pkd2−/− mouse inner medullary duct (mIMCD3) cells were a kind gift from the Kleene lab (University of Cincinnati). NIH3T3 cells were obtained from American Type Culture

after serum re-addition, after the indicated treatments. Primary cilia were visualized with Arl13b (green) (D). Serum-induced deciliation rates are shown as best-fits with 95% confidence limits in indicated treatments (n = 3) (E). Scale bars: 5 μm. Data are presented as means ± SEM. Two-way ANOVA with Holm–Sidak's multiple comparisons test. *P < 0.05, ***P < 0.001. **(F, G, H, I)** Graphic representation of the constructs encoding FLAG-GEMININ or FLAG-NDE1-GEMININ (F). Expression of the constructs was induced by doxycycline induction at the 0 h time point (G). Representative images from Pkd1-null NIH3T3 cells at 24 h after serum re-addition, transfected with the indicated constructs. dsRED⁺ cells (red) were evaluated for the presence of primary cilia via Arl13b staining (green) (H). Serum-induced deciliation rates are shown as best-fits with 95% confidence limits in indicated treatments (n = 3) (I). Scale bars: 5 μm. Data are presented as means ± SEM. Two-way ANOVA with Holm–Sidak's multiple comparisons test. ***P < 0.001.
Source data are available for this figure.

Collection. MEFs were maintained in DMEM (4.5 g/l glucose, L-glutamine, and sodium pyruvate) supplemented with 1% penicillin–streptomycin, 1% nonessential amino acids and 10% FBS. mIMCD3 cells were cultured in a 1:1 mixture of DMEM and Ham's F-12, supplemented with 10% FBS. NIH3T3 cells were maintained in DMEM containing 10% bovine calf serum.

## Reagents

Paraformaldehyde (Cat. no. J19943-K2), EdU (Cat. no. C10337), Prolonged Diamond DAPI (Cat. no. P36966), Trizol (Cat. no. 15596018), and LTX Reagent (Cat. no. 15338100) were purchased from Thermo Fisher Scientific. Nutlin-3 (Cat. no. 3984) and pifithrin-$\alpha$ hydrobromide (Cat. no. 1267) were purchased from Tocris. 4-hydroxytamoxifen was purchased from Sigma-Aldrich (Cat. no. H6278).

## Plasmids

lentiCRISPRv2-puro was a gift from Brett Stringer (plasmid # 98290; http://n2t.net/addgene:98290; RRID:Addgene_98290; Addgene). pDZ Flag USP28 was a gift from Martin Eilers (plasmid # 15665; http://n2t.net/addgene:15665; RRID:Addgene_15665; Addgene). pX330 p53 was a gift from Tyler Jacks (plasmid # 59910; http://n2t.net/addgene:59910; RRID:Addgene_59910; Addgene). pcDNA3 flag p53 was a gift from Thomas Roberts (plasmid # 10838; http://n2t.net/addgene:10838; RRID:Addgene_10838; Addgene). pcDNA5-FRT/TO-eGFP-53BP1 was a gift from Daniel Durocher (plasmid # 60813; http://n2t.net/addgene:60813; RRID:Addgene_60813; Addgene).

## CRISPR–CAS9 gene editing

For the generation of NIH3T3$^{Pkd1-KO}$ and NIH3T3$^{Pkd2-KO}$ clones, NIH3T3 cells were transfected with a lentiviral vector (lentiCRISPRv2-puro) encoding Cas9, resistance to puromycin, and single guide RNA (sgRNA) sequences specific for Pkd1 (5′-AGGGCCGAGTCTGCGCATCC-3′) or Pkd2 (5′-AAGTGCTGAAGTCATCGACC-3′). After transfection, the cells underwent serial dilutions and were plated under puromycin selection (1 $\mu$g/ml) for 3 wk. After selection, single colonies were isolated and expanded. The resulting clones were evaluated for deletion of the genes of interest by Sanger DNA sequencing and Western blotting.

## Immunoprecipitation

Wild-type or Pkd1-null NIH3T3 cell lysates were incubated with two $\alpha$-PKD1 antibodies (7e12; Santa Cruz and E4; PKD Baltimore Center) overnight to immunoprecipitate PKD1. The antigen–antibody complexes were then incubated with Protein G Sepharose beads for 3 h in 4°C. Immunoprecipitates were analyzed for the presence of PKD1 by Western blot.

## Indirect immunofluorescence on cultured cells

Cells were initially seeded on glass coverslips at a density of ~30,000 cells/well in 24-well plates. To induce synchronization in G1/G0 and primary cilia formation, cells were partially or completely deprived of serum for 48 h. Specifically, MEFs were cultured with OptiMEM low-serum medium, mIMCD3 cells were cultured with

DMEM/F12 medium containing 0.5% FBS and NIH3T3 cells were kept under DMEM medium with no serum. For induction of deciliation, the cells were restimulated with their respective full growth media (described above). After serum restimulation, the cells were fixed with 4% paraformaldehyde in PBS, pH 7.5, at different time points. Visualization of primary cilia was performed as described before (Maskey et al, 2015). Briefly, the fixed cells were permeabilized with 0.15% Triton X-100 in PBS for 90 s and blocked with 3% goat serum/0.15% Triton X-100 in PBS for 1 h. Then, the fixed cells were incubated overnight at 4°C with primary antibodies detecting Arl13b (1:500; Proteintech), $\gamma$-tubulin (1:500; Invitrogen), $\gamma$-H2AX (1:200; Cell Signaling), USP28 (1:100; Proteintech), and 53BP1 (1:1,000; Novus Biologicals). Next, the cells were washed 3× with PBS and incubated with Alexa Fluor–conjugated secondary antibodies (1:2,000; Invitrogen) for 2 h at 4°C protected from light. The samples were then washed 3× with PBS and mounted with Diamond DAPI (Thermo Fisher Scientific) to counterstain the nuclei. Images were obtained by an Olympus FV1000 confocal microscope and processed with ImageJ software for measurement of ciliary length.

## Rescue and siRNA experiments

For the deciliation rescue experiments, wild-type or mutant NIH3T3 cells were co-transfected, using LTX (Invitrogen), with a plasmid expressing GFP and one of the following plasmids expressing a) empty vector, b) human PKD1, c) PKD1$^{S99I}$ d) human PKD2, e) PKD2$^{D511V}$, or f) PKD2$^{Y684del}$. Next, the cells underwent synchronization by serum starvation and induced to deciliate and reenter the cell cycle by serum. GFP-positive cells were evaluated for the presence of primary cilia. p53 was targeted using a plasmid expressing single guide RNA targeting mouse p53 (Addgene) in wild-type or mutant NIH3T3 cells. Usp28 and 53bp1 knockdown experiments were performed using SMARTpool: ON-TARGETplus Usp28 siRNA (Dharmacon, Cat. no. L-065548-01-0005) and SMARTpool: ON-TARGETplus 53bp1 siRNA (Dharmacon, Cat. no. L-042290-01-0005), respectively. The cells were co-transfected, using LTX, with GFP and either one of the siRNA constructs and analyzed for the presence of primary cilia after serum-induced deciliation. Expression plasmids bearing USP28, 53BP1, or p53 cDNAs were co-transfected with GFP into wild-type NIH3T3 cells using LTX and GFP-positive cells were evaluated for the presence of primary cilia after serum induction.

## Mice and kidney tissue processing

All procedures were performed according to requirements of the Institutional Animal Care and Use Committee. In this study, three types of C57Bl/6-background mice were used: a) Pkd1$^{f/f}$, as wild-type littermates, b) UbcCre$^{ERT2}$;Pkd1$^{f/f}$, and c) KspCre$^{ERT2}$;Pkd1$^{f/f}$. For the deletion of Pkd1, nursing dams were intraperitoneally injected with 4-hydroxytamoxifen (4-OHT; Sigma-Aldrich) diluted in corn oil, at 100 $\mu$g per gram of mouse body weight, from postnatal days 2–6 (P2–P6). 1 d before euthanasia, the pups were intraperitoneally injected with EdU in PBS (20 $\mu$g of EdU per gram of mouse body weight) and were euthanized after 24 h. Before euthanizing, total body weight of the mice was measured. The kidneys were removed from each mouse and weighed. One of the kidneys was flash-frozen in liquid nitrogen and stored in −80°C. Frozen kidneys were later

lysed and fractionated into nuclear and cytosolic fractions using a Cytoplasmic & Nuclear Protein Extraction Kit (Cat. no. P504L; 101Bio). The other kidney was split in half, stored in 4% PFA in PBS overnight, then in 10% sucrose in PBS overnight, mounted on cryomolds with frozen section compound (Leica) for 30 min and then flash-frozen on dry ice and stored at −80°C. 10-$\mu$m-thick frozen kidney sections were prepared at the University of Oklahoma Health Sciences Center (OUHSC) Stephenson Cancer Center Histology Core and were used for indirect immunofluorescence staining.

### Indirect immunofluorescence on kidney tissue samples

All experiments were performed using the Thermo Scientific Sequenza staining system (Cat. no. 73310017), based on a protocol kindly provided by Dr. Pazour (UMass, MA). Briefly, frozen kidney sections were initially incubated in a Tris-buffered saline-Tween (TBST) bath for 5 min for the removal of frozen section compound and then mounted on coverplates (Cat. no. 7211017; Thermo Fisher Scientific) and incubated with blocking buffer (4% non-immune goat serum/0.1% cold water fish skin gelatin/0.1% Triton X-100 in TBST) for 30 min. EdU-positive cells were then visualized according to the manufacturer's instructions. If kidney segment staining was desired, endogenous biotin was blocked with a Streptavidin/Biotin Blocking Kit (Vector). After washing 3× with TBST, endogenous mouse IgG was blocked with Fab antimouse IgG prepared from secondary antibody host for 2 h (1:10 dilution in 0.1% fish skin gelatin in TBST; Jackson Immunoresearch). Then, the sections were incubated with primary antibody against acetylated-$\alpha$-tubulin (611b, 1:5,000; Sigma-Aldrich), $\gamma$-H2AX (1:200; Cell Signaling) or GEMININ (1:100; Proteintech) overnight at 4°C (diluted in 0.1% fish skin gelatin in TBST). For visualization of kidney segments, the samples were incubated with biotinylated Dolichos Biflorus Ag-glutinin (Vector) to mark collecting ducts or biotinylated Lotus Tetragonolobus Lectin (Vector) to mark proximal tubules. After 5× washing in TBST, the sections were incubated with Alexa Fluor–conjugated secondary antibodies (1:1,000) for 1 h in RT protected from light, washed 5× with TBST, incubated in a TBST bath for 15 min and then in a TBS bath for 15 min. Finally, sections were mounted with Diamond DAPI and visualized with confocal microscopy.

### Immunoblotting

Cell lysates were obtained in a lysis solution containing 1% Triton X-100, 150 mM NaCl, 10 mM Tris–HCl at pH 7.5, 1 mM EGTA, 1 mM EDTA, 10% sucrose, a protease inhibitor cocktail (Roche Applied Science), and phosphatase inhibitors 0.2 mM $Na_3VO_4$ and 1 mM NaF at 4°C for 30 min. Cell or tissue lysates were separated with SDS–PAGE. Antibodies were used against p-Smad2 465/467 (1:1,000; Cell Signaling), GAPDH (1:4,000; Genetex), p53 (1:1,000; Cell Signaling or 1:500; Millipore), Dishevelled-2 (1:1,000; Cell Signaling), INPP5E (1:1,000; Proteintech), NDE1 (1:1,000; Proteintech), USP28 (1:1,000; Proteintech), 53BP1 (1:1,000; Novus Biologicals), $\beta$-tubulin (1:1,000; Santa Cruz), $\beta$-actin (1:1,000; Santa Cruz) PKD1 (E8, 1:1,000; PKD Baltimore Center), and PKD2 (G-20, 1:1,000; Santa Cruz). Densitometric quantification was performed with the Licor Image Studio software.

### Signaling pathway screening

Activity of 45 different cellular signaling pathways was measured following the manufacturer's instructions (Cat. no. CCA-901L-12; Qiagen). Briefly, wild-type and $Pkd1$- or $Pkd2$-null NIH3T3 cells, or wild-type and $Pkd2$-null mIMCD3 cells were seeded for reverse transfection in a 96-well plate containing immobilized reporter plasmids. Every pair of wells contained plasmids expressing firefly luciferase in a manner dependent on activity of a specific cellular pathway. Every well also contained a plasmid that constitutively expressed Renilla luciferase, to account for differences in transfection efficiency. Reverse transfection was mediated by Attractene (Cat. no. 301005; Qiagen). Full growth medium was replaced 24 h after transfection, and cells were lysed in lysis buffer (Cat. no. E194A; Promega) 48 h after transfection. Cell lysates were incubated with Stop & Glo substrates (Dual-Luciferase Reporter Assay System, Cat. no. E1960; Promega), and luciferase activity was measured in a Synergy Neo2 Reader (Biotek). All firefly luciferase values were normalized to Renilla luciferase values, and then all mutant sample values were normalized to wild type.

### Quantitative RT-PCR (qRT-PCR)

RNA was extracted and purified from wild-type or $Pkd1$-null NIH3T3 cells at different time points after serum restimulation using TRIzol reagent (Thermo Fisher Scientific). RNA was reverse-transcribed to cDNA, and samples were amplified by qPCR. mRNA levels of the genes of interest were normalized to wild type at 0 h, via the ΔΔCt method. Primers used for qPCR were $Usp28$ Fw: 5′-AGTTGGGCT-GAAAAATGTTGGC-3′, $Usp28$ Rv: 5′-TCAAGGATGTTCTGTGGCAGG-3′, $53bp1$ Fw: 5′-CACGCCAGTTTTCACTCCTG-3′, $53bp1$ Rv: 5′-TGATGGTTCTTCCAGAC TTGGT-3′, $Tbp$ Fw: 5′-TCTACCGTGAATCTTGGCTGT-3′, and $Tbp$ Rv: 5′-GTCCGTGGCTCTCTTATTCTCA-3′.

### Statistics

Software GraphPad Prism 8.3 was used for all statistical analyses described in the text. Quantitative results that required comparisons between groups were subjected to statistical analysis using $t$ test for two groups or one- or two-way ANOVA followed by an appropriate ad hoc test.

## Supplementary Information

## Acknowledgements

We thank Drs. S Kleene for providing the IMCD3 cells lacking PKD2, G Pazour and M Hinsdale for providing tissue immunofluorescence labeling protocols, and W Berry for advice on CRISPR/Cas9 experiments. We thank the Histology and Immunohistochemistry Core of the University of Oklahoma Health Sciences Center (OUHSC) Stephenson Cancer Center for histology and tissue sectioning, supported by P30CA225520 and P20GM103639. This work was

supported by DK59599 and DK117654 (National Institutes of Health) and the Presbyterian Health Foundation and the John S. Gammill Endowed Chair in Polycystic Kidney Disease (University of Oklahoma Health Sciences Center) to L Tsiokas. V Gerakopoulos was supported by a predoctoral fellowship from the American Society of Nephrology.

## Author Contributions

V Gerakopoulos: conceptualization, data curation, formal analysis, funding acquisition, investigation, and writing—original draft, review, and editing.
P Ngo: methodology.
L Tsiokas: conceptualization, data curation, supervision, funding acquisition, investigation, project administration, and writing—original draft, review, and editing.

## Conflict of Interest Statement

The authors declare that they have no conflict of interest.

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
