## [Reviewer comments · Life Science Alliance]

Loss of Polycystins suppresses deciliation via the activation of the centrosomal integrity pathway

Vasileios Gerakopoulos, Peter Ngo, Leonidas Tsiokas

DOI: 10.26508/lsa/202000750

Corresponding author(s): Prof. Leonidas Tsiokas (OUHSC)

Review timeline:

Submission Date:	2020-04-21
Editorial Decision:	2020-04-22
Revision Received:	2020-05-30
Editorial Decision:	2020-06-22
Revision Received:	2020-07-01
Accepted:	2020-07-03

Transaction Report:

No Peer Review Process File is available with this article, as the authors have chosen not to make the review process public in this case.

Re: Life Science Alliance manuscript #LSA-2020-00750-T

Prof. Leonidas Tsiokas
University of Oklahoma Health Sciences Center
Cell Biology
975 NE 10th str
BRC1/262
Oklahoma City, OK 73104

Dear Dr. Tsiokas,

Thank you for transferring your manuscript entitled "Loss of Polycystins suppresses deciliation via the activation of the centrosomal integrity pathway" to Life Science Alliance. The manuscript was assessed by expert reviewers at another journal, and the editors transferred those reports to us with your permission.

The reviewers who evaluated your work elsewhere appreciated your data. However, they would have expected more support for the proposed mechanism and further reaching insight.

We would like to invite you to submit a revised version of your work to us, based on the reports already at hand. The data on the cell cycle (mainly figure 3) as well as the FBW7 data should get excised to address the reviewer concerns regarding the conclusiveness of these data. The reviewer comments pertaining to the other data should get addressed, and a full point-by-point response should get provided.

In our view these revisions should typically be achievable in around 3 months. However, we are aware that many laboratories cannot function fully during the current COVID-19/SARS-CoV-2 pandemic and therefore encourage you to take the time necessary to revise the manuscript to the extend requested above. We will extend our 'scoping protection policy' to the full revision period required. If you do see another paper with related content published elsewhere, nonetheless contact me immediately so that we can discuss the best way to proceed.

Thank you for this interesting contribution to Life Science Alliance. We are looking forward to receiving your revised manuscript.

Sincerely,

B. MANUSCRIPT ORGANIZATION AND FORMATTING:

Re: Life Science Alliance manuscript #LSA-2020-00750-TR

Prof. Leonidas Tsiokas
OUHSC
Cell Biology
975 NE 10th Street
Oklahoma City, OK 73104

Dear Dr. Tsiokas,

Thank you for submitting your manuscript entitled "Loss of Polycystins suppresses deciliation via the activation of the centrosomal integrity pathway" to Life Science Alliance. Your revised manuscript was now assessed by two of the original reviewers from the other journal, and their comments are appended to this letter.

As you can see, both reviewers find the revised manuscript has improved, but also indicate that further discussion and textual clarification would be required before they can recommend publication of the study. Therefore I would invite you to address the remaining comments from the two reviewers, and to include the geminin data in the manuscript as requested by Reviewer #1.

Please also take care of the following minor revisions:

- please add callouts for Fig. S3 A,B in the manuscript text
- please list only 10 authors and et al. in the reference list
- We can only publish figures that adhere to our guidelines, please revise accordingly (figure 4 & figure 6 span two pages at the moment)

Thank you for this interesting contribution to Life Science Alliance. We are looking forward to receiving your revised manuscript.

Sincerely,

Reilly Lorenz
Editorial Office Life Science Alliance
Meyerhofstr. 1
69117 Heidelberg, Germany
t +49 6221 8891 414
e contact@life-science-alliance.org
www.life-science-alliance.org

B. MANUSCRIPT ORGANIZATION AND FORMATTING:

RE: Life Science Alliance Manuscript #LSA-2020-00750-TRR

Prof. Leonidas Tsiokas
OUHSC
Cell Biology
975 NE 10th Street
Oklahoma City, OK 73104

Dear Dr. Tsiokas,

Thank you for submitting your Research Article entitled "Loss of Polycystins suppresses deciliation via the activation of the centrosomal integrity pathway". It is a pleasure to let you know that your manuscript is now accepted for publication in Life Science Alliance. Congratulations on this interesting work.

DISTRIBUTION OF MATERIALS:

Again, congratulations on a very nice paper. I hope you found the review process to be constructive and are pleased with how the manuscript was handled editorially. We look forward to future exciting submissions from your lab.

Sincerely,

Reilly Lorenz
Editorial Office Life Science Alliance
Meyerhofstr. 1

69117 Heidelberg, Germany
t +49 6221 8891 414
e contact@life-science-alliance.org
www.life-science-alliance.org